# Disease Differentiation and Monitoring of Anti-TNF Treatment in Rheumatoid Arthritis and Spondyloarthropathies

**DOI:** 10.3390/ijms22147389

**Published:** 2021-07-09

**Authors:** Katarzyna Bogunia-Kubik, Wojciech Wojtowicz, Jerzy Swierkot, Karolina Anna Mielko, Badr Qasem, Joanna Wielińska, Renata Sokolik, Łukasz Pruss, Piotr Młynarz

**Affiliations:** 1Laboratory of Clinical Immunogenetics and Pharmacogenetics, Hirszfeld Institute of Immunology and Experimental Therapy, Polish Academy of Sciences, 53-114 Wroclaw, Poland; katarzyna.bogunia-kubik@hirszfeld.pl (K.B.-K.); joanna.wielinska@hirszfeld.pl (J.W.); 2Department of Bioorganic Chemistry, Wroclaw University of Technology, 50-370 Wroclaw, Poland; wojciech.wojtowicz@pwr.edu.pl (W.W.); karolina.mielko@pwr.edu.pl (K.A.M.); badr.qasem@pwr.edu.pl (B.Q.); lukasz.pruss@pwr.edu.pl (Ł.P.); 3Department of Rheumatology and Internal Medicine, Wroclaw Medical University, 50-556 Wroclaw, Poland; jerzy.swierkot@umed.wroc.pl (J.S.); renata.sokolik@umed.wroc.pl (R.S.); 4Ardigen, 30-394 Krakow, Poland

**Keywords:** metabolomics, rheumatoid arthritis, ankylosing spondylitis, psoriatic arthritis, biological treatment outcome

## Abstract

Rheumatoid arthritis (RA), ankylosing spondylitis (AS), and psoriatic arthritis (PsA) are comprehensive immunological disorders. The treatment of these disorders is limited to ameliorating the symptoms and improving the quality of life of patients. In this study, serum samples from RA, AS, and PsA patients were analyzed with metabolomic tools employing the 1H NMR method in combination with univariate and multivariate analyses. The results obtained in this study showed that the changes in metabolites were the highest for AS > RA > PsA. The study demonstrated that the time until remission or until low disease activity is achieved is shortest (approximately three months) for AS, longer for RA and longest for PsA. The statistically common metabolite that was found to be negatively correlated with the healing processes of these disorders is ethanol, which may indicate the involvement of the gut microflora and/or the breakdown of malondialdehyde as a cell membrane lipid peroxide product.

## 1. Introduction

A complex of factors may induce immune responses and lead to the development of autoimmune diseases. Long-term inflammation contributes to the pathological state and is associated with organ-specific and systemic disorders, such as rheumatoid arthritis (RA), ankylosing spondylitis (AS), and psoriatic arthritis (PsA), which are of considerable interest to researchers at present [1]. RA, AS, and PsA are autoimmune diseases associated with changes in the joints and spine due to long-term inflammation, and these diseases are associated with significant reductions in patients’ quality of life.

Rheumatoid arthritis is a chronic, inflammatory autoimmune disease that affects approximately 1% of the population. RA is described as synovial inflammation and joint destruction that leads to significant disability and early mortality. The etiology of RA is multifarious and has not been fully elucidated to date, although genetic and environmental factors have been implicated in disease development [2].

Many inflammatory processes involving various immune cells, cytokines, chemokines, proteases, and matrix metalloproteinases play critical roles in the inflammatory cascade of the joint environment, leading to clinical impairment and RA [3]. Even though therapy with TNF (tumor necrosis factor)-alpha inhibitors constitute a breakthrough in RA management, this treatment results in no improvement in approximately 30% of cases. The reasons for anti-TNF therapy failure have not been determined to date. Identifying predictive biomarkers is critical to optimizing benefits to patients and reducing the cost of treatment, as well as minimizing the considerable adverse effects related to therapy. Thus, the importance of personalized therapy is becoming increasingly clear [4].

Ankylosing spondylitis is a chronic, progressive spinal inflammatory arthritis with a diverse clinical presentation that belongs to the spondyloarthropathies (SpA) group. The estimated prevalence of axial spondyloarthritis is similar to that of rheumatoid arthritis. Chronic inflammatory back pain is the leading symptom of the disease. Other musculoskeletal manifestations include arthritis and enthesitis [5]. Inflammation processes associated with AS can cause bone erosion, new bone formation, and ankylosis occurring in the spine, which leads to severe pain and stiffness and a reduced spinal mobility [6]. The primary goal of treating patients with AS is to maximize health-related quality of life through the control of symptoms and inflammation, prevention of progressive structural damage, preservation/normalization of function, and social participation [7,8].

Psoriatic arthritis is classified into the SpA group and shares a genetic and clinical background with them [9]. PsA is characterized by clinical phenotypes involving the peripheral and axial skeleton, as well as various manifestations, including dactylitis, entheses, joints, and nails [10]. PsA coexists with skin disease and develops in up to 30% of psoriasis patients. In addition, other rheumatic diseases often lead to limited function and reduced quality of life [11,12]. Both similarities and differences were observed between these diseases. Spinal involvement is characteristic of AS in 40% of PsA patients but not RA patients. Rheumatoid factor and anti-cyclic citrullinated peptide antibodies are commonly absent in PsA patients, which is in contrast to patients diagnosed with RA. HLA-B*27 is present in approximately 90% of AS patients, but in patients with psoriatic arthritis, a positive test result may also appear [13]. PsA patients are also characterized by the presence of other class I molecules [14].

Metabolomics is a powerful, rapid, and comprehensive tool for exploring changes in the concentration of metabolites. Therefore, metabolomics may be useful for evaluating disease susceptibility biomarkers, diagnostic approaches, treatment outcomes, and pharmaceutical dosages [15]. This method enables the analysis of molecules smaller than 1.5 kD by liquid chromatography–mass spectrometry (LCMS), gas chromatography–mass spectrometry (GC–MS), or nuclear magnetic resonance (NMR) spectroscopy [16].

The pathogenesis of all three rheumatoid diseases described above is associated with genetic and environmental factors. There has also been research, conducted both by our group and others, indicating the role of metabolomics studies [17,18].

A previous study performed on early RA patients show that changes in serum metabolites may be predictor factors of MTX efficiency, after 24 weeks of treatment [19]. However, the application of metabolomics in predicting clinical response to anti-TNF therapy is still limited. Most studies are focused on rheumatoid arthritis patients and incorporate different techniques to perform metabolic profiling [20]. Previous data obtained on CD4 + T cells showed that MTX and infliximab have minimal toxicity at clinically relevant concentrations in RA patients. In this case, global metabolic changes have been analyzed by LC–MS or GC–MS [21]. Cuppen et al. also used LC–MS analysis to investigate the baseline serum of RA patients receiving TNF inhibitors [22].

Moreover, ^1^H-NMR-based studies were successfully employed to identify anti-TNF treatment predictors. Kapoor et al. screened the urine metabolome of RA and PsA patients and showed changes in profiles between baseline and 12 weeks of anti-TNF therapy. They also found that histamine, glutamine, xanthurenic acid, and ethanolamine discriminate the biological drug response in RA patients. In addition, the RA baseline metabolic profile was correlated with the magnitude of the one-year change in the disease activity score in 28 joints. What is more, the urine metabolome differs between etanercept and infliximab treatment in responders. Additionally, altered levels of isoleucine, leucine, valine, alanine, glutamine, tyrosine, glucose, and 3-hydroxybutyrate were noted in good responders after six months of treatment [23].

Moreover, Takahashi et al., using capillary electrophoresis–time-of-flight mass spectrometry (CE–TOFMS), identified betonicine, glycerol 3-phosphate, N-acetylalanine, hexanoic acid and taurine as serum biomarkers able to predict a response in RA patients receiving anti-TNF therapy [24]. Recently, Ou et al. proved that serum metabolites were also correlated with AS and TNF inhibitor treatment [25].

These results motivated us to analyze the metabolomic profiles of rheumatic disorders. During the present study, metabolomics was used to characterize and compare unique disease-associated metabolites in the serum of RA, AS, and PsA patients treated with anti-TNF drugs. The metabolites of RA, AS, and PsA patients were analyzed before the induction of biological treatment and at subsequent time points—i.e., 3 and 6 months after therapy initialization. In the end, our main goal was to identify potential biomarkers that may be employed in the future to elucidate the pathophysiology and treatment outcomes of these diseases.

## 2. Results

### 2.1. Response to Treatment

Among patients with RA, after 3 months of treatment, low disease activity was achieved in three (11%) patients, and disease remission was achieved in five patients (19%). After 6 months of therapy, these results were 13 (50%) and nine patients (35%), respectively. No response according to EULAR was observed in four patients (15%) after 6 months (Appendix A).

In the group treated for AS, 13 patients (45%) and one patient (3%) achieved low disease activity (BASDAI between 3 and 4) in the third and sixth month of therapy, respectively. In the same time frame, 13 (45%) and 26 patients (90%) achieved remission, respectively. No response was observed in three patients (10%) after 3 months and two patients (7%) after 6 months of therapy.

Among patients with PsA after 3 months of treatment, seven patients (30%) had low disease activity, and three (13%) had remission. After 6 months of treatment, these results were four (17%) and 11 (48%), respectively. No response was observed in 13 patients (57%) after 3 months and in eight patients (35%) after 6 months.

### 2.2. Rheumatoid Arthritis (RA) Patients

#### Monitoring of Treatment Response in RA Patients

Metabolomic profiles were analyzed and compared in serum samples collected from RA patients before anti-TNF treatment (BF) and 3 months (3M) and 6 months (6M) after initialization of the treatment (Figure 1 and Appendix A). The first multivariate analysis (Figure 1) performed by PCA (principal component analysis) exhibited large scattering of patients without any grouping trends among PC1 and PC2. The patients’ comparison between each time period by PLS-DA (partial least-squares discriminant analysis) showed the differences between BT vs. 3M and BT vs. 6M; however, the model 3M vs. 6M did not pass the validation by CV-ANOVA (cross-validated residuals of analysis of variance) test (Appendix A, Table 1). The VIP (variable importance in projection) score comparison between each analyzed time point demonstrates the differences between each state by the importance of single metabolites in the model variance explanation (Appendix A). The quantified resonance signals were tested by ANOVA, and their dispersions are shown in Figure 2. The evaluation of changes during whole treatment progression identified eight metabolites in which seven metabolites—2-oxoisocapoate, 3 methyl-2-oxovalerate, alanine, glutamine, propylene glycol, tryptophan, and tyrosine—were increasing, and only one—ethanol—was decreasing (Figure 2).

### 2.3. Ankylosing Spondylitis (AS) Patients

#### Monitoring of Treatment Response in AS Patients

The multivariate analysis showed the same trend between treatment progressions in the studied AS patient groups, as was observed for RA (Figure 3 and Appendix A, Table 2). Significant differences were observed between BT vs. 3M and BT vs. 6M. Nonetheless, each time period was characterized by different metabolites (Table 2, Figure 4), which may reflect a considerably more altered metabolism in AS patients than in RA patients during the treatment. Several metabolites exhibit similar variability, such as 2-oxoisocaproate, 3-methyl-2-oxovaletarte, ethanol, glutamine, propylene glycol, and tryptophan (Appendix A). An increasing trend between BT and two other treatment time points was observed for creatine, histidine, leucine, phenylalanine, and UNK_11 (unknown). Four metabolites—citrate, formate, sn-G3P, and UNK_15—first increased and later decreased through treatment progression. Six metabolites—acetone, ethanol, isobutyrate, UNK_10, UNK_2, and UNK_8—distinctly decreased over the course of the treatment (Appendix A, Figure 4).

### 2.4. Psoriatic Arthritis (PsA) Patients

#### Monitoring of Treatment Response for PsA Patients 

Anti-TNF therapy did not reflect the greatest changes in PsA treatment. The MVA did not show any significant grouping trends (PCA) (Figure 5), and none of the calculated PLS-DA models passed the CV-ANOVA test (Appendix A, Table 3). Nonetheless, the univariate analysis exhibited decreased levels of four metabolites—acetate, ethanol, UNK_2, and UNK_8 (Appendix A, Figure 6).

### 2.5. Metabolomic Profile of Patients at Two Time Points: Before Treatment and after 6M Treatments

#### Comparison of AS, RA and PsA Patients

Metabolomic profiles of patients with various rheumatic diseases were compared by analyzing serum samples collected from 26 patients with RA, 29 patients with AS and 23 patients with PsA before initialization of biological therapy. All patients presented with active, advanced disease.

This analysis showed that RA, AS, and PsA were characterized by different levels of metabolites before treatment (Appendix A). However, the unsupervised multivariate analysis did not show any grouping trends among PC1 and PC2. A significant PLS-DA model was obtained in the RA vs. AS individual case comparison (Table 4, Figure 7). The RA vs. PsA and AS vs. PsA comparisons did not pass the CV-ANOVA validation test.

The univariate analysis revealed 12 resonance signal relative integrals that were common for all three pathological units: ethanol, isoleucine, leucine, UNK_7, valine, proline, alanine, histidine, UNK_4, sn-G3PC, UNK_8, and UNK_3 (Table 5). These resonance signals were compared for the studied groups at the BT time point.

To verify possible dissimilarities between threatened patients of the studied groups, the comparison between serum samples originating from patients after 6 months of treatment outcome was evaluated.

Subsequently, ANOVA or Kruskal–Wallis tests identified seven metabolites, and two unknown signals were determined to be significant for the comparison of RA, AS, and PsA after 6M (Table 6, Figure 8). Assessment of important changes in metabolite levels highlighted the visible relative integral decrease commonly observed in the PsA group. Most decreased levels were observed in creatine, UNK_16, lysine, sn-3GP, UNK_14 and acetate. The AS entity exhibited the most prominent increase in the levels of ethanol and glutamate. Moreover, acetate and sn-3GP exhibited differentiating possibilities only between AS and PsA diseases. Analogs were visible between RA and PsA in lysine, creatine, UNK_14, and UNK_16.

## 3. Discussion

The possibility of monitoring treatment progression and the quality of response in terms of patient quality of life seems to be vital for future treatment improvement. At present, metabolomics is a well-defined and well-utilized scientific discipline. This approach enables quantitative and qualitative assessment of the composition of a variety of different biological materials. With the use of modern chemical analytical platforms—that is, mass spectrometry (MS) and nuclear magnetic resonance spectroscopy (NMR) together with statistical and chemometrics analysis—metabolomics became a powerful tool to collect multivariate biological data from a single biosample for cohort studies. Together with current data analysis methods, this approach enables us to investigate the “needle in a haystack” approach to detect even the slightest variation in the studied biological material.

In our previous study, this approach was employed to analyze and compare the metabolomic profiles of females with RA and female controls [17]. This analysis enabled us to identify 12 metabolites important for the discrimination of both patient and control groups. In this study, the metabolomic serum approach was also employed for monitoring rheumatoid arthritis treatment. We observed that after treatment, patients did not move toward the healthy controls but rather formed a separate group. Moreover, the differences in metabolites between patients with various IL-17 genotypes (rs2275913 and G-197A) were determined to affect RA progression and response to anti-TNF-α treatment [26].

In the present study, metabolomics analyses were employed to collect and verify changes among low-molecular-weight compounds in 78 patients suffering from RA, AS, and PsA. Patients were analyzed at three time points: before and 3 and 6 months after initialization of treatment with a biologic agent. There are important differences between both previous and present studies. The former study focused on 20 female RA patients analyzed at two time points (before and 3 months after TNFi induction) compared with the control group, whereas in the current study, both male and female patients with three rheumatic diseases were included. Moreover, different sample preparations were applied based on methanol protein precipitation to maximize the biological information obtained from 1D ^1^H NMR spectra. Furthermore, MVA models were calculated based on relative integrals and not on binned spectra.

### 3.1. Comparison of Three Rheumatic Diseases (RA vs. AS vs. PsA) before Therapy Induction

Among all multivariate analyses, only one RA vs. AS comparison passed the validation test, showing the differences in metabolite contents between these two disease units. This result could suggest that these differences between RA and AS may be caused by either the differences in molecular basis between these two rheumatic disorders or, most likely, the different contribution to the variables from all time points, which reflects the treatment impact on the metabolic fingerprint of the patients.

ANOVA or the Kruskal–Wallis test demonstrated that in most cases, the level of metabolites between studied units was at the lowest value in RA.

These results suggest that the most significant differences between RA and AS metabolomics profiles might also be associated with the clinical pictures of these diseases. AS and PsA belong to the same group of so-called spondyloarthropathies.

PsA may be clinically oligoarthritis, may resemble RA, or may have an axial form that resembles AS. Among the studied patients, the most common PsA was diagnosed with oligoarthritis or polyarticular forms (87%). The picture of changes in the synovial membrane of inflamed joints may often be similar in patients with RA and PsA, but on the other hand, changes in bones are different. In RA patients, joint destruction (erosions) is observed, while in PsA, destruction and new bone formation (erosions and enthesophytes) are observed.

Patients suffering from RA and PsA before starting biological drugs were previously treated with classic DMARDs, mainly methotrexate (65%), and patients with AS were treated primarily with NSAIDs (only nine were treated with MTX).

RA does not involve the spine but the peripheral joints, while AS is an axial spondyloarthropathy that primarily affects the spine and sacroiliac joints. Moreover, both groups of patients (RA and AS) differed with respect to the percentage of males and females. RA primarily affects women, while AS primarily affects men, and this relationship is also clearly visible in our analyzed groups (Appendix A). Nevertheless, it appears that RA and AS can be distinguished based on the observed differences in metabolic profiles. These results may suggest that metabolomics could serve as a severe diagnostic tool.

### 3.2. Characterization of RA Group during the Treatment

Analyzing the calculated PLS-DA models clearly showed that each of the treatment time intervals exhibited alterations in reference to the before treatment group. However, the period 3M vs. 6M seems to be on the marginal value of significance, which highlighted the clear need for further expanding the size of the studied group. This finding may suggest that each period of therapy significantly changes the subject’s metabolism, and monitoring this change is possible. The treatment is distinctly reflected by the changes in metabolite levels (Appendix A), as they are important at different time points. Regarding the paired samples comparison (Figure 2), the amino acid (Ala, Trp, Tyr, and Glu) levels are increasing, which is in agreement with previously published data and reflects the overall trend that the amino acid pool is low in RA [27]. 2-Oxoisocaproate and 3-methyl-2-oxovalerate are two ketoanalogs of ketoleucine and ketoisoleucine. Both of these metabolites are neurotoxins and metabotoxins; therefore, the finding that their levels increase with the patient’s improving condition is difficult to explain. However, these metabolites were recognized to be downregulated in RA patients [28], where they were associated with cartilage destruction caused by decreasing of amino acids levels [29]. The increase in these metabolites may be caused by increasing leucine and isoleucine levels and turnover, which, according to previously published data, are decreased in comparison to healthy people [30,31]. Additionally, in this study, both amino acids were increased (Table 3). These amino acids can balance the energy demand during increased energy consumption and lack of energy from other sources [3].

Propylene glycol is a known “concomitant” to originate from cosmetics and was found to be a component of the serum metabolome [32]. However, the extraction of propylene glycol during sample preparation cannot be completely excluded. Notably, the level of this compound increases with time, which may indicate that the equilibrium between its metabolization release and storage is changed during therapy. However, the increasing level of this compound may be associated with less need for energy substrates, as propylene glycol can be converted to lactate [33].

This hypothesis can be supported by increasing the levels of alanine and glutamine, which can both funnel the tricarboxylic acid cycle (TCA) pathway, alanine to pyruvate and glutamine by glutamate to a-ketoglutarate. However, alanine and glutamine metabolism are mutually related [34]. Increased glutamine was determined to be a marker between responders and non-responders to etanercept treatment [23]. Notably, TCA production can be accompanied by increases in citrate and lactate together. This phenomenon can be caused by glycolysis/gluconeogenesis energy sources (amino acids) shifting toward high pyruvate biosynthesis (Table 3). However, high lactate production has a completely opposite trend in the published data [35].

Notably, the ethanol level was decreased, and this metabolite can be associated with the consumption of alcoholic beverages, but in our opinion, this association may be a very “rough” assumption. Similarly, the microbial activity in serum samples can be excluded due to sample storage at −80 °C [36]. Therefore, the presence of autogenerated endogenous ethanol should be considered. This finding could be closely related to the activity of the gut metabolome, where the presence of the filamentous fungus [37,38,39] Candida albicans can modulate autoethanol production. On the other hand, ethanol is closely related to acetaldehyde generated from pyruvate, threonine, deoxyribose-5-phosphate, phosphoethanolamine, and alanine [40].

Acetaldehyde can also be the product of malondialdehyde breakdown as the product of cell disintegration and accompanying membrane lipid peroxidation [41].

Interpretation of PLS-DA models VIP scores (>1.00) and univariate statistical analysis highlights the metabolites set with high impact in discrimination possibilities.

### 3.3. Characterization of AS Group during the Treatment

The PLS-DA models showed that the BT time point differs from 3M and 6M, while 3M vs. 6M did not exhibit significant changes, most likely due to continuously occurring healing processes and/or more rapid replies to drug intervention. Appendix A shows the changes in metabolite levels, while considering the metabolites of paired samples, it is clear that treatment with AS caused considerably more metabolic changes than treatment with RA, where only a few of the same metabolites were changed. Among these metabolites, Gln and Trp increased, but the stabilization effect was already seen after 3M (Trp) or showed a slight drop (6M-Gln). In addition, tryptophan was found to be downregulated in the plasma of AS and RA patients, which can be strongly associated with the activity of indoleamine 2,3-dioxygenase (IDO), which transforms tryptophan to kynurenine [24,35] and was suggested to be an indicator of disease progression. According to a prior hypothesis, two other amino acids—His and Leu—were increasing, and their levels after 3M were already equilibrated. Observed earlier than amino acids, keto analogs first demonstrated an increasing trend after 3M, and a stabilization effect was subsequently observed. The reverse trend in Phe was reported in the literature data in comparison to the obtained results, where this metabolite was upregulated in AS patients, whereas its dipeptide Phe-Phe was downregulated [42]. However, Phe was found to be statistically important in RA and showed the same trend as in this study [31].

Propylene glycol and ethanol showed the same trend as that observed for RA. The increasing trend of citrate can be associated with an accelerated TCA cycle, which reversibly correlates with keto body acetone levels, which may be a sign of decreased ketoacidosis. Formic acid is a naturally occurring metabolite in the serum metabolome; in this study, its level first increased and then decreased. The presence of formic acid can be modulated by internal metabolism or by gut microbiota [17]. In our previous study, the level of formic acid was decreased in female RA patients undergoing anti-TNF therapy [17,26]. The next metabolite that can be associated with the human microbiome is isobutyrate, which decreased significantly with the sampling time. This serum short-chain amino acid is the product of gut valine degradation [43,44] and can be related to the lipid profile [45]. Creatine metabolism is associated with Ser, Gly Thr, Arg, and Pro metabolism; however, none of these compounds were observed to be significantly changed [KEGG]. However, supplementation with creatine can attenuate muscle loss [46]. Glycerol 3-phosphate is a product of glucose breakdown, and the intermediate substrate for lipid metabolism can also be obtained by the reduction of dihydroxyacetone-3-phosphate [47]. Therefore, the increase in G3P levels may be caused by accelerated lipolysis, rather than glycolysis. The observed changes in the abovementioned metabolite levels confirm that, at least in part, the same biochemical pathways are unblocked and that some new pathways are also activated. These results suggest that the keto analogs, amino acids, ethanol, creatine, and isobutyrate may reflect the treatment outcome and serve as potential biomarkers.

### 3.4. Characterization of the PsA Group during the Treatment

The metabolic analysis clearly demonstrated that this disease entity requires long-term treatment. The positive therapy results were visible only after 6 months (Appendix A, Table 3); however, the smallest changes between metabolites were observed among RA and AS patients. The univariate analysis also identified only two metabolites that were changed, namely, acetate and ethanol. Both metabolites exhibited decreasing levels. Ethanol is a biomarker that appears to be associated with rheumatoid-based inflammation in all investigated entities, while short-chain fatty acid acetate has been determined to be significant for the first time. Short-chain fatty acids were found to be important in RA mice, where acetate was increased in serum RA mice, while the therapeutic effect of butyrate, which ameliorates the immune-systemic response, was demonstrated [48].

### 3.5. Relationship between Identified Metabolites and Inflammation/Disease Activity Parameters (CRP, DAS28, VAS, BASDAI); Bioinformatic Analysis

Monitoring changes in DAS28, CRP (C reactive protein), and VAS (visual analog scale) in RA showed significant improvement after 6 months, while after 3 months, partial improvement in parameters was observed. Analyzing the AS parameters, CRP, VAS, and BASDAI improved after 3 months. These findings confirm that 3M treatment has the same effect as 6M treatment but can retain the treatment effect. The parameters of the disease activity of PsA, DAS28, CRP, and VAS were not satisfactory in 35% of patients, even after 6M.

All these findings reflect the treatment efficacy, showing that fewer treated PsA patients are characterized by low disease activity and achieve remission compared to two other diseases, especially AS.

In the literature, DAS28 was negatively associated with histidine and was well correlated with its changes [49]. Our studies have shown that the increase in statistically important amino acids corresponds to a decrease in all inflammation parameters. However, amino acids, such as proline, isoleucine, tryptophan, valine, arginine, ornithine, kynurenine, 4-hydroxyproline, and leucine, were positively correlated with CRP [22]. The negative correlation across all disease entities showed that ethanol, which decreased, clearly reflects the therapeutic efficiency. Thus, it seems that ethanol levels decrease during anti-TNF drug administration and are associated with more favorable outcomes of treatment with this biologic agent.

Notably, a study by Azizov et al. suggested that moderate alcohol consumption may be a consistent protective factor for the development of autoimmune diseases in mice [50].

The results described earlier by Jansson et al. have shown that this effect of ethanol could be mediated by (i) downregulation of leukocyte migration and (ii) upregulation of testosterone secretion, with the latter leading to decreased NF-κB (nuclear factor kappa-light-chain-enhancer of activated B cells) activation [51].

The relationship between anti-TNF treatment efficacy and ethanol levels observed in the present study may also be indirectly affected by the decreased NF-κB expression associated with genetic variability of the gene encoding this transcription factor. Indeed, in vitro functional studies [49] have suggested that the presence of the deletion may be associated with diminished expression of the gene. Notably, we have previously observed a significantly more efficient response to anti-TNF-α treatment in RA patients carrying the deletion within the NF-κB1 gene [52,53]. Moreover, we have also found some associations between polymorphisms within the TLR4-encoding gene and RA stage, as well as response to anti-TNF-α therapy [53].

Thus, our studies indicate a beneficial effect of using a biologic agent to treat RA patients carrying the deletion allele associated with lower NF-κB expression. As suggested by Jannson et al., the diminished activity of this transcription factor can mediate the anti-inflammatory and anti-destructive properties of ethanol in mice with collagen-induced arthritis. In addition, the previously reported relationships with TLR4 genetic variants (receptors with LPS as a ligand) may serve to confirm, as suggested by the results of the present metabolomic study, the involvement of the microbiome in the development of rheumatic diseases [53].

The changes in all perturbed metabolites were subjected to bioinformatics analysis consisting of modules including perturbed compounds, associated enzymes, perturbed pathways, and reactions in which assigned compounds participate (interactive link, Appendix A) [54]. Additionally, the meta-analysis was performed, showing the most perturbed biochemical pathways—see Appendix A [55].

## 4. Materials and Methods

### 4.1. Patients with Rheumatic Diseases

Altogether, 78 patients with rheumatic diseases were investigated, including patients with RA (*n* = 26), AS (*n* = 29), and PsA (*n* = 23) hospitalized at the University Hospital in Wrocław in the Department of Rheumatology and Internal Medicine of Wroclaw Medical University. All patients gave their informed consent for inclusion before they participated in the study. The study was conducted in accordance with the Declaration of Helsinki, and the protocol was approved by the Wroclaw Medical University Ethics Committee (identification code KB-625/2016, 29 December 2016).

The exclusion criteria to participate in the study were as follows: clinically significant impairment of hepatic and renal function, coexistence of connective tissue diseases, infections with hepatotropic viruses, or infections resistant to therapy, ongoing history of cancer or uncontrolled diabetes, alcohol abuse, pregnancy, breastfeeding, unwillingness to cooperate, or insufficient clinical records.

RA patients met the 2010 European League Against Rheumatism (EULAR)/American College of Rheumatology (ACR) classification criteria. The inclusion criteria were the following: age over 18 years, Caucasian origin, high disease activity (DAS28) ≥ 5.1) before initiating biologic agent therapy, no response to at least two disease-modifying anti-rheumatic drugs (DMARDs), and a complete medical history. The disease activity of RA patients was determined using the DAS28 score based on four components, including the number of swollen and tender joints, C-reactive protein (CRP) level, erythrocyte sedimentation rate (ESR) and patient’s global assessment of general health expressed on a visual analog scale (VAS, mm). Except for DAS28 measurement, anti-cyclic citrullinated peptide antibodies (anti-CCP) and rheumatoid factor (RF) levels were determined. The patients were stratified into three subgroups depending on disease activity: high (DAS28 > 5.1), moderate (3.2 < DAS28 ≤ 5.1), and low (DAS28 ≤ 3.2), and their responses to anti-TNF therapy after 3 and 6 months were assessed by the European League Against Rheumatism (EULAR) criteria. A response was considered good when reduction of the DAS28 score value (ΔDAS28) > 1.2 and post-treatment DAS28 ≤ 3.2, as moderate when ΔDAS28 > 1.2 and post-treatment DAS28 > 3.2, or 0.6 < ΔDAS28 ≤ 1.2 and post-treatment DAS28 ≤ 5.1. Finally, no response was assumed when ΔDAS28 ≤ 0.6 or 0.6 < ΔDAS28 ≤ 1.2 and post-treatment DAS28 > 5.1.

AS patients were certified according to the modified New York criteria for AS and Assessment of Spondylo Arthritis international Society (ASAS) classification criteria for axial and peripheral spondyloarthritis (SpA) and the Bath Ankylosing Spondylitis Disease Activity Index (BASDAI). The following inclusion criteria were applied: age over 18 years, Caucasian origin, high disease activity (BASDAI ≥ 4) before initiation of biologics, resistance to treatment with at least two nonsteroidal anti-inflammatory drugs (NSAIDs) for at least four weeks at maximum doses (if there were no contraindications) and full medical history. The disease activity of AS patients was estimated using the BASDAI, a number of swollen and tender joints, values of CRP and ESR, global health evaluation provided by a patient (VAS), spinal mobility and assessment of extra-articular manifestations. The disease activity was considered to be high (BASDAI ≥ 4), moderate (3 ≤ BASDAI <4), or low (BASDAI < 3). The ASAS/EULAR criteria were employed to assess the clinical outcome after 3 and 6 months of anti-TNF treatment.

Significant improvement was defined as a reduction in BASDAI (ΔBASDAI ≥ 2.0), good outcome as ΔBASDAI ≥ 2.0 and BASDAI < 3.0 at the endpoint, moderate response as ΔBASDAI ≥ 2.0 and BASDAI ≥ 3.0 at the endpoint, and no response as ΔBASDAI < 2.0.

PsA patients were diagnosed according to the Classification Criteria for Psoriatic Arthritis (CASPAR criteria). Patients included in the study were characterized by subsequent criteria: age over 18 years, Caucasian origin, a complete medical history and physical examination, failure with treatment with at least two disease-modifying anti-rheumatic drugs (DMARDs) for four months in peripheral form and at least two nonsteroidal anti-inflammatory drugs for at least four weeks at maximum doses (if there were no contraindications) in axial form, and the presence of active disease prior to the initiation of anti-TNF therapy.

Disease activity in PsA patients was calculated using the DAS28 score, Disease Activity Score (DAS) or modified criteria for disease activity according to the Psoriatic Arthritis Response Criteria (PsARC) in peripheral form of psoriatic arthritis and BASDAI in axial form. High disease activity was defined in axial form as BASDAI ≥ 4 and in peripheral form as DAS28 ≥ 5.1 or DAS > 3.7. The PsARC is based on counts of swollen and tender joints, physician global assessment of disease activity (zero- to five-point Likert scale), and questionnaires for levels of pain and spinal mobility.

TNF-alpha inhibitors were administered to the RA and AS patients according to standard protocols: 40 mg of adalimumab (ADA) administered subcutaneously every other week; 50 mg of etanercept (ETA) administered subcutaneously every week; 400 mg of certolizumab pegol (CER pegol) administered subcutaneously at weeks 0, 2 and 4, then 200 mg every second week thereafter; golimumab (GOL) administered subcutaneously 50 mg once a month and on the same day each month.

Additionally, AS patients received infliximab (INF) 3 mg/kg of body weight, given as intravenous infusions at weeks 0, 2, and 6, and every 8 weeks thereafter. PsA was treated analogously to RA and AS with ADA, ETA, CER pegol, and GOL.

Patient serum samples were collected at three time points: before treatment (BT) and 3 and 6 months (3M and 6M) after the initialization of anti-TNF drug administration.

The patients’ characteristics are summarized in Appendix A.

### 4.2. Metabolic Studies

#### 4.2.1. 1D ^1^H NMR Measurements (CPMG) of Patient Serum Sample Preparation

The serum samples were stored at −80 °C. Before preparation, serum samples were thawed in an ice bath, 300 mL was obtained, and it was mixed with 600 mL methanol (Merck KGaA, Darmstadt, Germany). The samples were then mixed for 1 min and placed at −20 °C for 20 min. After that procedure, the samples were centrifuged (30 min, 11,000 rpm, 4 °C), and 700 µL of clarified upper phase was transferred into new tubes. The solvent was evaporated in a vacuum centrifuge (40 °C, 1500 rpm for 4 h). In the next step, 600 µL of PBS buffer (0.5 M, pH = 7.0, TSP (ARMAR AG, Döttingen, Swisserland) = 0.03 mM, 20% D_2_O (ARMAR AG, Döttingen, Swisserland) was added to each sample and mixed for 3 min. Subsequently, 550 µL of each sample was transferred into 5-mm NMR tubes (5SP, Armar Chemicals, Leipzig, Germany). Until the measurement was taken, the samples were stored at 4 °C.

#### 4.2.2. NMR Measurements and Preprocessing

All NMR spectra were recorded using a Bruker 600 MHz AVANCE II spectrometer and by using the CPMG pulse sequence (cpmgpr1, *Bruker* notation) with following parameters relaxation delay of 3.5 s, acquisition time of 2.72 s, 128 scans, time domain of 65 k, and spectral width of 20 ppm. The line broadening was set at 0.3 Hz. All NMR spectra were referenced to the TSP resonance (δ = 0.000 ppm). Phased correction and baseline correction were corrected manually. Spectra were normalized to the constant sum of the TSP signal. The alignment of resonance signals was performed using the correlation optimized warping (COW) and icoshift functions implemented in MATLAB environment (v R2019a, Mathworks Inc., Natick, MA, USA) [56,57]. The relative integral of NMR measured metabolites was obtained as a sum of data points of the nonoverlapping resonances or a cluster of partly overlapping resonances from the data matrix consisting of 60,474 data points for each spectrum in n-dimension. The third quartile value of the noise region was subtracted from the calculated relative integral to decrease the influence on the final values.

#### 4.2.3. Metabolites Identification

The metabolite resonances were identified based on chemical shifts and the results of STOCSY [58] analysis and according to online databases (Biological Magnetic Resonance Data Bank [59] and Human Metabolome Data Base [60]) and assignments published in the literature based on ^1^H NMR chemical shifts.

#### 4.2.4. Univariate Analysis

The calculations for univariate statistics were performed on the relative integral of metabolite values. For normality verification, the Shapiro–Wilk test was performed. All statistical tests were calculated at a significance level of α = 0.05. All measured metabolites were checked with Pearson’s rho to verify possible interactions between known small molecular compounds and unidentified compounds. The equality of variances was tested with Levene’s test. The comparison based on the overtime treatment response was tested dually depending on the number of patients who dropped out. If the number of dropouts was below three, then observations without pairs were removed from the dataset. If the number of dropouts was above three, the testing was based on partially paired data with use of *t*-test with the pooled Fisher’s method for *p* values [61]. The false discovery rate (FDR) based on the Benjamini Hochberg procedure was applied for the tested metabolites. Multiple group comparisons were performed on nonpaired data using ANOVA and Tukey HSD or Kruskal–Wallis and Dunn–Sidak correction tests depending on the data distribution. The graphical representation and percentage difference were prepared on a general dataset without consideration of the fulfillment of paired samples.

#### 4.2.5. Multivariate Data Analysis

The MVA analysis was performed on integral relative metabolites. All the relative integral variables were scaled to unit variance (UV). The sample order in the data matrix was randomized. Data analysis was performed using two methods: unsupervised principal component analysis (PCA) for natural clustering of samples and supervised partial least squares discriminant analysis (PLS-DA) for identifying variables responsible for biological differences. The MVA data visualization marked an ellipse with Hotelling’s T2 range (95%). Partial least squares discriminant analysis with a sevenfold cross-validation procedure was employed to determine variation between studied groups. The reliability of the PLS-DA models was assessed by cross-validation analysis of variance (CV ANOVA) at a significance level of α = 0.05. The most important variables in discrimination between comparisons, were selected based on the variable importance in projection (VIP) values with a cutoff value of 1.00.

#### 4.2.6. Bioinformatic Analysis

To obtain Kyoto Encyclopedia of Genes and Genomes (KEGG) IDs from the resulting list of metabolites, the most recent compounds downloaded from the KEGG API http://rest.kegg.jp/list/compound (accessed on 19 October 2020) were mapped to the corresponding metabolites. After unification of input, we employed the FELLA package to build a KEGG-based hierarchical representation of human biochemistry (pathways, modules, enzymes, reactions, and metabolites). First, we retrieved the tabular KEGG data for humans (T01001, Release 96.0+/12–13 December 20) to build the knowledge graph. Later, we mapped the list of input metabolites to the internal representation, creating an enriched object, and we subsequently ran the propagation algorithm-diffusion method (undirected heat diffusion model) to score graph nodes. Additionally, the parametric z-score was computed using normality approximations for statistical normalization [54].

## Figures and Tables

**Figure 1 ijms-22-07389-f001:**
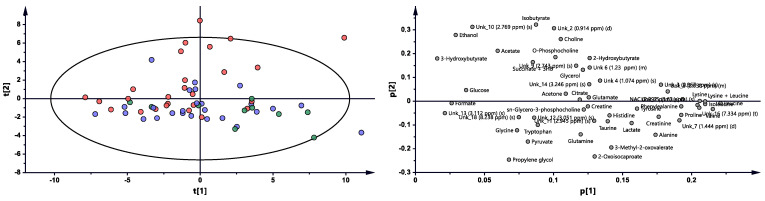
PCA model plot and corresponding loading plot for RA patients studied at three time points. Red—RA patients before treatment; blue—3 months after initialization of anti-TNF treatment; green—6 months after treatment with TNFi.

**Figure 2 ijms-22-07389-f002:**
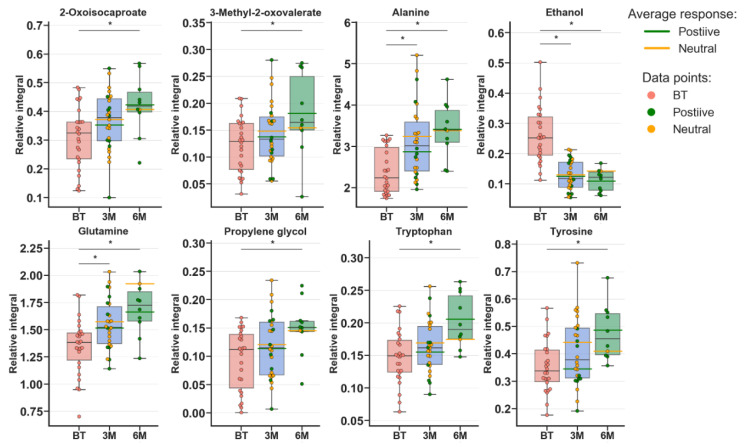
Boxplots for metabolites with VIP scores above 1.00 and statistical importance after *p* value adjustment (*q* < 0.05). Red bars—RA patients before treatment; blue bars—3 months after initialization of anti-TNF treatment; green bars—6 months after treatment with TNFi. Whiskers—1.5 × interquartile range (IQR); bar—average; box—range between first quartile (Q1) and third quartile (Q3). Green line—average positive response for treatment; yellow line—average neutral response for treatment. Pink circle—data point for before treatment; red circle—data point for no information about response; green circle—data point for positive response for treatment; yellow circle—neutral response for treatment. * *q* value < 0.05 (precise values in Appendix A).

**Figure 3 ijms-22-07389-f003:**
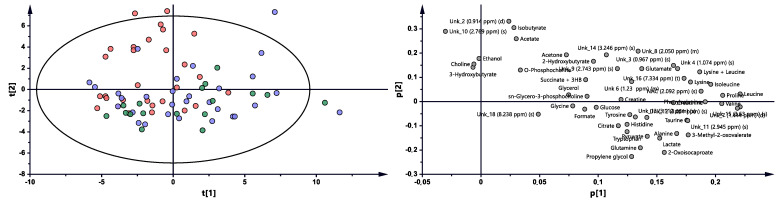
PCA model plot and corresponding loading plot for AS patients studied at three time points. Red—AS patients before treatment; blue—3 months after initialization of anti-TNF treatment; green—6 months after treatment with TNFi.

**Figure 4 ijms-22-07389-f004:**
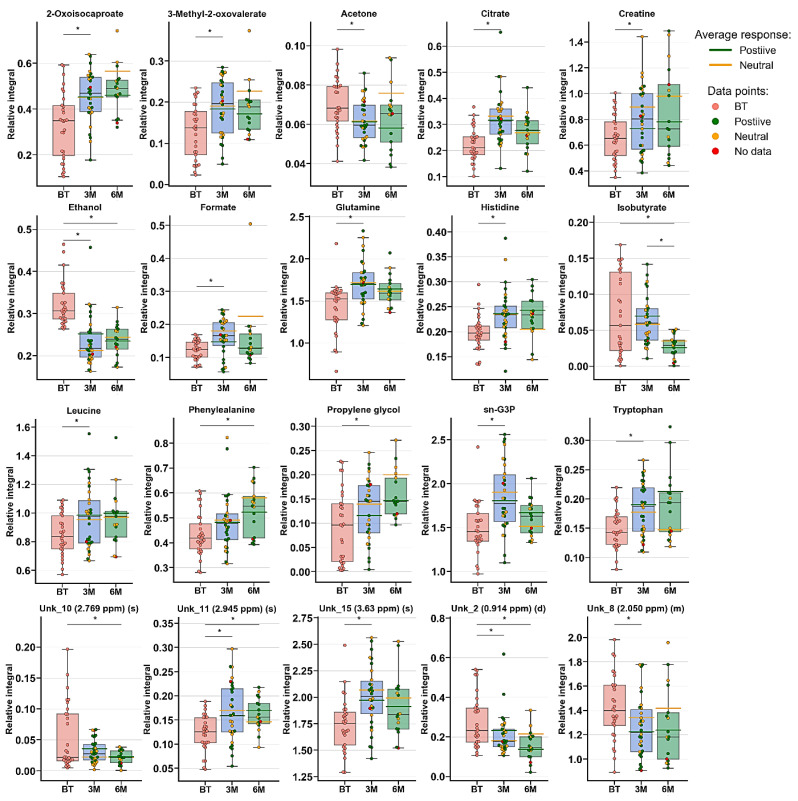
Boxplots for metabolites with VIP scores above 1.00 and statistical importance after *p* value adjustment (*q* < 0.05). Red bars—AS patients before treatment; blue bars—3 months after initialization of anti-TNF treatment; green bars—6 months after treatment with TNFi. Whiskers—1.5 × interquartile range (IQR); bar—average; box—range between first quartile (Q1) and third quartile (Q3). Green line—average positive response for treatment; yellow line—average neutral response for treatment. Pink circle—data point for before treatment; red circle—data point for no information about response; green circle—data point for positive response for treatment; yellow circle—neutral response for treatment. * *q* value < 0.05 (precise values in Appendix A).

**Figure 5 ijms-22-07389-f005:**
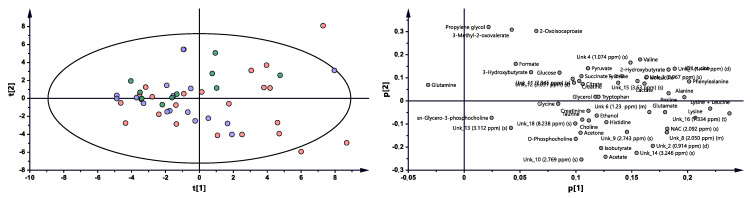
PCA model plot and corresponding loading plot for PsA patients studied at 3 months after initialization of anti-TNF treatment; green—6 months after treatment with TNFi.

**Figure 6 ijms-22-07389-f006:**
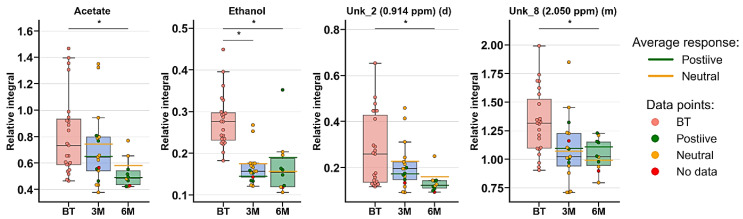
Boxplots for metabolites with VIP scores above 1.00 and statistical importance after p-value adjustment (*q* < 0.05). Red bars—PsA patients before treatment; blue bars—3 months after initialization of anti-TNF treatment; green bars—6 months after treatment with TNFi. Whiskers—1.5 × interquartile range (IQR); bar—average; box—range between first quartile (Q1) and third quartile (Q3). Green line—average positive response for treatment; yellow line—average neutral response for treatment. Pink circle—data point for before treatment; red circle—data point for no information about response; green circle—data point for positive response for treatment; yellow circle—neutral response for treatment. * *q* value < 0.05 (precise values in Appendix A).

**Figure 7 ijms-22-07389-f007:**
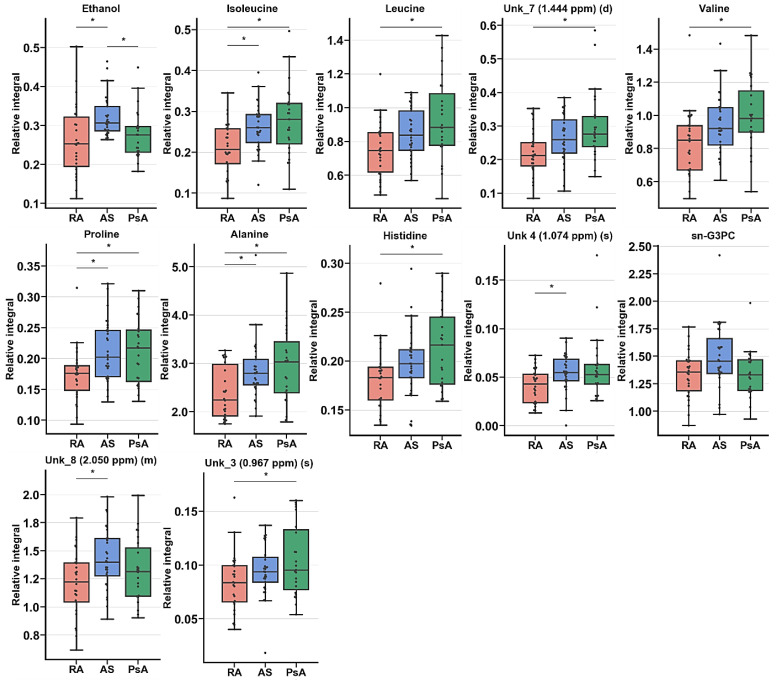
Boxplot for statistically important metabolites in ANOVA or Kruskal–Wallis test Figure 1; 5 × interquartile range (IQR); bar—average; box—range between first quartile (Q1) and third quartile (Q3). * Metabolites important in Tukey’s HSD or Dunn–Sidak test.

**Figure 8 ijms-22-07389-f008:**
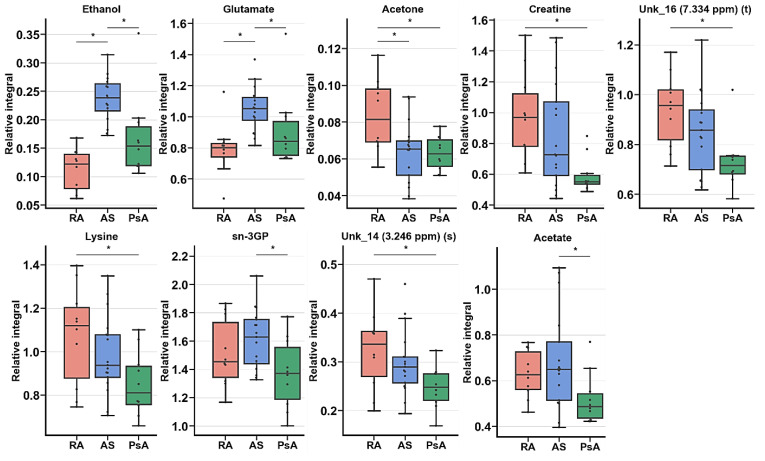
Boxplot for statistically important metabolites for RA, AS, and PsA in ANOVA (In appearance order 2,3,8,12) or Kruskal–Wallis (1,4–7,9–11) test for RA, AS, and PsA groups of patients before treatment. Whiskers—1.5 × interquartile range (IQR); bar—average; box—range between first quartile (Q1) and third quartile (Q3). * Metabolites important in Tukey’s HSD (2,3,8,12) or Dunn–Sidak test (1,4–7,9–11).

**Table 1 ijms-22-07389-t001:** The multivariate analysis model summary for RA patient treatment influence.

Comparison	Model Type	PC/LV	N =	R^2^X (cum)	R^2^Y (cum)	Q^2^ (cum)	CV-ANOVA *p* Value
BT vs. 3M vs. 6M	PCA	5	62	0.627	−	−	−
BT vs. 3M	PLS-DA	3	52	0.482	0.726	0.494	5.23 × 10^−6^
BT vs. 6M	PLS-DA	4	36	0.594	0.902	0.577	9.25 × 10^−4^
3M vs. 6M	PLS-DA	2	36	0.384	0.563	0.235	7.21 × 10^−2^

**Table 2 ijms-22-07389-t002:** The multivariate analysis models summary for AS patients’ treatment influence.

Comparison	Model Type	PC/LV	N =	R^2^X (cum)	R^2^Y (cum)	Q^2^ (cum)	CV-ANOVA *p* Value
BT vs. 3M vs. 6M	PCA-X	4	76	0.564	-	-	-
BT vs. 3M	PLS-DA	3	59	0.476	0.679	0.347	4.09 × 10^−4^
BT vs. 6M	PLS-DA	4	46	0.536	0.897	0.702	5.49 × 10^−8^
3M vs. 6M	PLS-DA	2	47	0.343	0.588	−0.0994	1.00

**Table 3 ijms-22-07389-t003:** The multivariate analysis models summary for PsA patients’ treatment influence.

Comparison	Model Type	PC/LV	N =	R^2^X(cum)	R^2^Y(cum)	Q^2^(cum)	CV-ANOVA *p* Value
BT vs. 3M vs. 6M	PCA-X	5	50	0.643	−	−	−
BT vs. 3M	PLS-DA	2	40	0.305	0.522	−0.00191	1
BT vs. 6M	PLS-DA	2	33	0.344	0.554	0.0492	0.999
3M vs. 6M	PLS-DA	2	27	0.249	0.604	−0.21	1

**Table 4 ijms-22-07389-t004:** The multivariate analysis models summary for RA, AS, PsA comparison.

Comparison	Model Type	PC/LV	N =	R^2^X(cum)	R^2^Y(cum)	Q^2^(cum)	CV-ANOVA *p* Value
RA vs. AS vs. PsA	PCA	2	78	0.417	-	-	-
RA vs. AS	PLS-DA	2	55	0.335	0.638	0.431	8.48 × 10^−6^
RA vs. PsA	PLS-DA	2	49	0.387	0.400	−0.028	1.00
AS vs. PsA	PLS-DA	2	52	0.282	0.457	−0.0513	1.00

**Table 5 ijms-22-07389-t005:** Summary of univariate analysis for RA, AS, PsA comparison at the BT time point.

Metabolite	*p* Value	Central Tendency		
		RA	AS	PsA
Ethanol ^(a)^	2.70 × 10^−3^	0.253	0.306	0.276
Isoleucine ^(b)^	2.90 × 10^−3^	0.212	0.262	0.279
Leucine ^(b)^	5.94 × 10^−3^	0.752	0.849	0.928
Unk_7 (1.444 ppm) (d) ^(a)^	8.04 × 10^−3^	0.213	0.260	0.276
Valine ^(a)^	8.15 × 10^−3^	0.852	0.921	0.982
Proline ^(a)^	9.69 × 10^−3^	0.176	0.202	0.217
Alanine ^(a)^	1.02 × 10^−2^	2.246	2.796	3.034
Histidine ^(b)^	1.34 × 10^−2^	0.183	0.198	0.214
Unk 4 (1.074 ppm) (s) ^(a)^	1.52 × 10^−2^	0.043	0.055	0.053
sn-G3PC ^(a)^	3.27 × 10^−2^	1.357	1.455	1.332
Unk_8 (2.050 ppm) (m) ^(a)^	4.14 × 10^−2^	1.224	1.399	1.316
Unk_3 (0.967 ppm) (s) ^(b)^	4.21 × 10^−2^	0.084	0.095	0.105

^(a)^ Parametric test (ANOVA), mean values; ^(b)^ nonparametric test (Kruskal–Wallis), median values.

**Table 6 ijms-22-07389-t006:** Summary of univariate analysis for RA, AS, PsA comparison after 6 months of anti-TNF treatment.

Metabolite	*p* Value	Central Tendency		
		RA	AS	PsA
Ethanol ^(b)^	1.46 × 10^−3^	0.122	0.239	0.154
Glutamate ^(b)^	9.12 × 10^−3^	0.802	1.054	0.844
Acetone ^(a)^	6.31 × 10^−3^	0.084	0.064	0.063
Creatine ^(b)^	9.67 × 10^−3^	0.968	0.728	0.552
Unk_16 (7.334 ppm) (t) ^(b)^	1.27 × 10^−2^	0.957	0.858	0.716
Lysine ^(a)^	3.90 × 10^−2^	1.074	0.986	0.852
sn-Glycero-3-phosphocholine ^(a)^	4.24 × 10^−2^	1.516	1.611	1.372
Unk_14 (3.246 ppm) (s) ^(b)^	4.33 × 10^−2^	0.336	0.290	0.248
Acetate ^(a)^ *	6.07 × 10^−2^	0.629	0.684	0.520

^(a)^ Parametric test (ANOVA), mean values; ^(b)^ nonparametric test (Kruskal–Wallis), median values; * statistically significant in post hoc analysis.

## Data Availability

The datasets generated and/or analyzed during the current study are available from the corresponding author on reasonable request.

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
