# Peer review of "Disease Differentiation and Monitoring of Anti-TNF Treatment in Rheumatoid Arthritis and Spondyloarthropathies"

_ijms, 2021, doi:10.3390/ijms22147389_

Round 1

Reviewer 1 Report

Serum samples from patients with Rheumatoid arthritis (RA), ankylosing spondylitis (AS), and psoriatic arthritis (PsA) were analyzed with metabolomic tools using the 1H NMR method in combination with univariate and multivariate analyses in this study. The paper highlighted that Ethanol was found to be statistically negatively correlated with the healing processes of these disorders, which could indicate the involvement of the gut microflora and/or the breakdown of malondialdehyde as a cell membrane lipid peroxide product. The paper will provide the audience with an informative reading, but the findings are mostly informative. A few points should be clarified and considered in order to further improve the manuscript:

Clarity and context: It would be worth adding an adequate literature to the introduction. The study's limitations should be discussed.

Originality and significance: The metabolomic assessment of serum samples of RA and PsA or similar diseases has been indicated by the literature. It should be extended to show what further information this paper could provide in order to make this study an interesting reading.

Validity: The authors showed that ethanol and glutamate had changed considerably. These findings should be validated and compared with the literature by another method / analysis.

Methodology and data analysis: Explicit details on the validation of peak assignments and the attempt to assign unknown peaks would be important. Bin width details and other relevant parameters of the data processing would be informative. Specific information should be given in the statistical analysis section. For example, line 269: the tests ANOVA or Kruskal-Wallis are indicated but no specific details are provided.

Results and Conclusions:  The findings are empirical and do not provide mechanistic details. Rigorous data analysis would also provide additional information. More evidence should be provided by the authors to back up their claims. Perhaps comparing metabolite level changes with RNA-Seq will provide additional evidence. Data analysis with MetaboAnalyst would be very informative. Authors should also discuss the subjects' nutritional status.

Language: There are grammatical and typographical errors. It is recommended that the manuscript be proofread.

References: The use of a reference appears to be appropriate.

Author Response

Dear Reviewer,

Thank you for all your efforts in reviewing of our manuscript the comments, suggestion and remarks we are aware that is to make our data more relevant for IJMS journal. Please find below our all answers to the raised questions.

Respond to Reviewer 1

Serum samples from patients with Rheumatoid arthritis (RA), ankylosing spondylitis (AS), and psoriatic arthritis (PsA) were analysed with metabolomic tools using the 1H NMR method in combination with univariate and multivariate analyses in this study. The paper highlighted that Ethanol was found to be statistically negatively correlated with the healing processes of these disorders, which could indicate the involvement of the gut microflora and/or the breakdown of malondialdehyde as a cell membrane lipid peroxide product. The paper will provide the audience with an informative reading, but the findings are mostly informative. A few points should be clarified and considered in order to further improve the manuscript:

  1. Clarity and context: It would be worth adding an adequate literature to the introduction. The study's limitations should be discussed.

Ad 1. This is done

  1. Originality and significance: The metabolomic assessment of serum samples of RA and PsA or similar diseases has been indicated by the literature. It should be extended to show what further information this paper could provide in order to make this study an interesting reading.

Ad 2. In respond to this point - the presented changes in metabolites are the valuable information which can indicate the progress of the diseases (as a bunch of biomarker simultaneously changed – profile) as well as the perturbed biochemical pathways which can be the targets for drug design. 

  1. Validity: The authors showed that ethanol and glutamate had changed considerably. These findings should be validated and compared with the literature by another method / analysis.

Ad 3. We expanded the introduction and added literature suggested by Reviewers

  1. Methodology and data analysis: Explicit details on the validation of peak assignments and the attempt to assign unknown peaks would be important. Bin width details and other relevant parameters of the data processing would be informative. Specific information should be given in the statistical analysis section. For example, line 269: the tests ANOVA or Kruskal-Wallis are indicated but no specific details are provided.

Ad 4. Table with precise information of spin system and chemical shift for variables which were used for calculation of univariate and multivariate analysis have been added as supplementary material (Table S1). For figure 8 description have been changed:

“Boxplot for statistically important metabolites in ANOVA (In appearance order 2,3,8,12) or Kruskal-Wallis (1,4-7,9-11) test for RA, AS and PsA groups of patients before treatment. Whiskers – 1.5×interquartile range (IQR); bar – average; box – range between first quartile (Q1) and third quartile (Q3). *Metabolites important in Tukey’s HSD (2,3,8,12) or Dunn-Sidak test (1,4-7,9-11).”

  1. Results and Conclusions:  The findings are empirical and do not provide mechanistic details. Rigorous data analysis would also provide additional information. More evidence should be provided by the authors to back up their claims. Perhaps comparing metabolite level changes with RNA-Seq will provide additional evidence. Data analysis with MetaboAnalyst would be very informative. Authors should also discuss the subjects’ nutritional status.

Ad 5. In general the suggested RNA-Seq require additional experiment, therefore we have use the bioinformatics tool to perform MetaboAnalyst analysis and we have introduced them to the supplementary data. Unfortunately, the nutritional status of the patients is unknown but will be included in the further studied

  1. Language: There are grammatical and typographical errors. It is recommended that the manuscript be proofread.

Ad 6. We have used the manuscript correction by  “American Journal Experts”

  1. References: The use of a reference appears to be appropriate.

Reviewer 2 Report

In this study, the authors aimed to analyze the metabolomics profiles of rheumatoid arthritis (RA), ankylosing spondylitis (AS), and psoriatic arthritis (PsA) before starting therapy with TNF inhibitors and at subsequent time points after therapy. Besides, they compared the unique disease-associated metabolites in the serum samples from patients with RA, AS, and PsA. Some comments need to be addressed as follows:   

  1. It would be better that the metabolic profiling of patients with RA, AS, and PsA, respectively could be mapped to available reference standards, and their biological pathways using the pathway analysis tool MetaboAnalyst (http://www.metaboanalyst.ca/).
  2. Given an importance of personized therapy, it is a critical issue to identify predictors (metabolites or metabolomic profiles) for good and non-response in patients receiving therapy with TNF inhibitors in patients with RA, AS, or PsA respectively.
  3. With the previous similar reports regarding metabolomics in patients with RA and AS receiving therapy with TNF inhibitors [Kapoor et al. Arthritis Rheumatology 2013; Chimenti et al. Cell Cycle 2013; Cuppen et a. PLoS One 2016; Xie et al. Clin Rheuamtol 2018; Takahashi et al. Rheumatology 2019; Ou et al. Frontiers Immunol 2021], the authors would have a literature review of metabolomics studies in patients with RA, AS, or PsA, and have the relevant discussion.
  4. AS can be divided into axial domain and peripheral arthritis domain. Similarly, PsA includes at least axial domain and peripheral arthritis either in oligoarthritis or polyarthritis. It would be interesting that the authors could present the differences in metabolites between different domains in each disease.
  5. Given that the used conventional synthetic disease-modifying antirheumatic drugs (csDMARDs) are different among patients with RA, AS, and PsA, the impact of different csDMARDs on the serum metabolites should be considered.
  6. Some typing errors, such as page 4, line 128: “single metabolites”; page 5, line 158: in PsA patients (AS patients?) under the subtitle 2.3. Ankylosing spondylitis (AS) patients, should be revised.

Author Response

Dear Reviewer,

Thank you for all your efforts in reviewing of our manuscript the comments, suggestion and remarks we are aware that is to make our data more relevant for IJMS journal. Please find below our all answers to the raised questions.

Respond to Reviewer 2

Comments and Suggestions for Authors

In this study, the authors aimed to analyze the metabolomics profiles of rheumatoid arthritis (RA), ankylosing spondylitis (AS), and psoriatic arthritis (PsA) before starting therapy with TNF inhibitors and at subsequent time points after therapy. Besides, they compared the unique disease-associated metabolites in the serum samples from patients with RA, AS, and PsA. Some comments need to be addressed as follows:   

  1. It would be better that the metabolic profiling of patients with RA, AS, and PsA, respectively could be mapped to available reference standards, and their biological pathways using the pathway analysis tool MetaboAnalyst (http://www.metaboanalyst.ca/).

  1. Given an importance of personized therapy, it is a critical issue to identify predictors (metabolites or metabolomic profiles) for good and non-response in patients receiving therapy with TNF inhibitors in patients with RA, AS, or PsA respectively.

  1. With the previous similar reports regarding metabolomics in patients with RA and AS receiving therapy with TNF inhibitors [Kapoor et al. Arthritis Rheumatology 2013; Chimenti et al. Cell Cycle 2013; Cuppen et a. PLoS One 2016; Xie et al. Clin Rheuamtol 2018; Takahashi et al. Rheumatology 2019; Ou et al. Frontiers Immunol 2021], the authors would have a literature review of metabolomics studies in patients with RA, AS, or PsA, and have the relevant discussion.

  1. AS can be divided into axial domain and peripheral arthritis domain. Similarly, PsA includes at least axial domain and peripheral arthritis either in oligoarthritis or polyarthritis. It would be interesting that the authors could present the differences in metabolites between different domains in each disease.

  1. Given that the used conventional synthetic disease-modifying antirheumatic drugs (csDMARDs) are different among patients with RA, AS, and PsA, the impact of different csDMARDs on the serum metabolites should be considered.

  1. Some typing errors, such as page 4, line 128: “single metabolites”; page 5, line 158: in PsA patients (AS patients?) under the subtitle 2.3. Ankylosing spondylitis (AS) patients, should be

Ad 1. Is done for each disease entities and inserted into supplementary data

Ad 3. The suggested literature are included

Ad 2, 4, 5 – GENERAL ANSWER

Reviewer suggestion are important and could give additional significant information in disease control and therapy. However, we do not considered these approaches in experimental design. Although we partially collected these information from patients but the studied groups don’t have precisely or reasonable balanced groups allowing for statistical correctness in data analysis.  The next experiment will be devoted for comparison of properly balanced groups.

Nonetheless, we prepared suggested analysis by Reviewer (separate pdf file) on collected data but in our opinion these comparisons should not be considered as a part of our manuscript.

Ad 2.

Markers for moderate and good treatment response are not possible to assess with meaningful statistical power, due to small sample size together with parried experimental rate and high patient dropout rate.

Ad 4.

Differences between AS Types (axial (1) vs peripheral (2)) no statistically significant PLS discrimination was observed in comparisons:

  1. AS Type 1, BT and AS Type 2, BT,
  2. AS Type 1, 3 months vs AS Type 2, 3 months,
  3. AS Type 1, 6 months vs AS Type 2, 6 months.

For AS different types analysis showed that in case of Type 1 the therapy influences patient metabolome as soon as 3 months, however for Type 2, first statistically significant differences in PLS-DA model is observed after 6 months.

Ad 5.

For therapy influence of antirheumatic drugs (MTX), only in AS and RA patients data have complete information. There were no statistical significant discriminant models for above data between checked comparisons:

  1. RA BT (MTX-Yes) and RA BT (MTX-No),
  2. RA 3 months (MTX-Yes) vs RA 3 months (MTX-No),
  3. RA 6 months (MTX-Yes) vs RA 6 months (MTX-No).
  4. AS BT (MTX-Yes) and AS BT (MTX-No),
  5. AS 3 months (MTX-Yes) vs AS 3 months (MTX-No),
  6. AS 6 months (MTX-Yes) vs AS 6 months (MTX-No).

Results for all additional analysis are attached in pdf presentation together with model parameters and workflow across Reviewer suggestions.

Ad 6. This is corrected, however we have left “single metabolites” which fits to the context of the text

Reviewer 3 Report

Manuscript ID: ijms-1251056

Title: Disease differentiation and monitoring of anti-TNF treatment in rheumatoid arthritis and spondyloarthropathies  

In this work, the authors aimed to investigate the disease-associated metabolites in the serum of rheumatoid arthritis (RA), ankylosing spondylitis (AS) and psoriatic arthritis (PsA) patients by nuclear magnetic resonance (NMR) spectroscopy, in combination with univariate and multivariate analyses. Moreover, the metabolomics profile was analyzed before and after (3 and 6 months) of biological treatment with anti-TNF drugs.

Comparing the three disorders, the changes in metabolites were highest in PsA and the time until remission or until low disease was also longest for PsA. Moreover, the common metabolite that was negatively correlated with the healing processes of these disorders was ethanol.

Overall the manuscript is well written, the research design is appropriated to address the main goal of the work, and the results are clearly presented. However, some details might be improved according to the comments mentioned below:

Major points: 

  1. I recommend the authors reformulate the Discussion section. In my point of view, the Discussion includes some results description that is most appropriate for the Results section. Moreover, I suggest highlighting the most relevant conclusions of the study in a final paragraph.

Minor points: 

  1. I suggest defining the abbreviations (e.g.  MVA, VIP, PCA, TSP, etc.) at the first time that appeared in the manuscript.
  2. In the Material and Methods, the authors include two topics of Multivariate data analysis (4.2.5 and 4.2.6). Why? Might the information be gathered in one only topic? 

Author Response

Dear Reviewer,

Thank you for all your efforts in reviewing of our manuscript the comments, suggestion and remarks we are aware that is to make our data more relevant for IJMS journal. Please find below our all answers to the raised questions.

Respond to Reviewer 3

Title: Disease differentiation and monitoring of anti-TNF treatment in rheumatoid arthritis and spondyloarthropathies  

In this work, the authors aimed to investigate the disease-associated metabolites in the serum of rheumatoid arthritis (RA), ankylosing spondylitis (AS) and psoriatic arthritis (PsA) patients by nuclear magnetic resonance (NMR) spectroscopy, in combination with univariate and multivariate analyses. Moreover, the metabolomics profile was analyzed before and after (3 and 6 months) of biological treatment with anti-TNF drugs.

Comparing the three disorders, the changes in metabolites were highest in PsA and the time until remission or until low disease was also longest for PsA. Moreover, the common metabolite that was negatively correlated with the healing processes of these disorders was ethanol.

Overall the manuscript is well written, the research design is appropriated to address the main goal of the work, and the results are clearly presented. However, some details might be improved according to the comments mentioned below:

Major points: 

  1. I recommend the authors reformulate the Discussion section. In my point of view, the Discussion includes some results description that is most appropriate for the Results section. Moreover, I suggest highlighting the most relevant conclusions of the study in a final paragraph.

Ad 1. We would like to thank the Reviewer for this recommendation. However, we would like to leave the current structure of the discussion, as it includes the analysis of the individual lesions of anti-TNF treatment in rheumatoid arthritis and spondyloarthropathies. The changed metabolites have been intentionally introduced in the discussion, which are also mentioned in the results.

Minor points: 

  1. I suggest defining the abbreviations (e.g.  MVA, VIP, PCA, TSP, etc.) at the first time that appeared in the manuscript.

      Ad 1. This is done

  1. In the Material and Methods, the authors include two topics of Multivariate data analysis (4.2.5 and 4.2.6). Why? Might the information be gathered in one only topic?

    Ad 2. Corrected

Round 2

Reviewer 1 Report

The current version of the manuscript and author's response to previous comments are acceptable.

Reviewer 2 Report

No any further comment.